# The Effects of Physicochemical Parameters on Analysed Soil Enzyme Activity from Alice Landfill Site

**DOI:** 10.3390/ijerph18010221

**Published:** 2020-12-30

**Authors:** Nontobeko Gloria Maphuhla, Francis Bayo Lewu, Opeoluwa Oyehan Oyedeji

**Affiliations:** 1Department of Chemistry, Faculty of Science and Agriculture, University of Fort Hare, Private Bag X1314, Alice 5700, South Africa; 2Department of Agriculture, Faculty of Applied Sciences, Cape Peninsula University of Technology, Wellington Campus, P.O. Box X8, Wellington 7655, Western Cape, South Africa; LewuF@cput.ac.za

**Keywords:** enzyme activity, soil pollution, physicochemical parameters, soil health

## Abstract

The soil pollution as a product of xenobiotics, industrial action, agricultural chemicals, or inappropriate disposal of waste can change the natural environment of soil indices and trigger life-threatening situations. Soil enzyme activity is the suitable bio-indicator or parameter for monitoring soil pollution due to their sensitivity that quickly responds to any soil disturbances. Also, they are known to play an essential role in maintaining soil health and a quality environment. This study aimed to determine the levels of enzyme activity in soil from polluted and unpolluted sites and study the relationship between the physicochemical properties and soil enzyme activity to manage soil pollution. Four selected enzymes (Urease, Invertase, Catalase, and Phosphatase) were examined for their activity from forty samples using the assay method for 24 h; the colorimetry spectrophotometer measured their activity. The obtained data revealed that Invertase activity was a predominant enzyme in all soil samples. Whereas, the urease activity had obtained in low amounts for all collection sites, especially on Site A1. Soil pH had discovered to range between 5.8 and 8.51, moisture content values recorded to vary from 0.12% to 9.09%, and soil organic carbon recorded to fluctuate between 0.08% and 1.54%. Urease and phosphatase activity correlated positively with all soil physicochemical properties except for moisture content, which correlated negatively (r = −0.297; *p* ≥ 0.05). The invertase activity negatively associated all soil physicochemical properties, excluding the moisture content that correlated positively and significantly with invertase activity. We noted that the dumpsite soil contains low enzyme activity levels, which might attribute to the type of waste disposed off. Also, only the phosphatase activity reported correlating positively with all examined physicochemical parameters entirely.

## 1. Introduction

Heavy metal pollution is a serious global environmental problem as it contributes to ecological disturbances. Toxic metals cause enzyme inactivation leading to changes in soil characteristics, limiting the productiveness and environmental functions. They can also enter the human body through the food chain and cause harm to people’s health. In plants, these pollutants prevent plant growth and result in an animal’s existence endangered. Furthermore, the burning of waste in the site to reduce garbage releases toxic particulate matter and smokes that cause respiratory complications and other health problems for people and other living organisms [1,2]. 

Therefore, determining the enzymes activity in the soil might indicate changes in the biological intensity process. The soil degradation levels, occurs due to the correlation with both the physical and chemical properties of soil. Soil enzymes are the important moderator and catalysts of significant roles in soil [3,4]. Other soil enzymes such as invertase, urease, phosphatase, arylsulfatase, and hydrolase play an essential role in nutrient cycling. They catalyse the cycling of nutrients such as carbon (C), nitrogen (N), phosphate (P), and sulphur (S) also organic matter (OM) decomposition in soils [5]. At present, soil enzymology is of practical significance because of industrial waste, agro-chemicals, heavy metals, and soil fertility management that can be measured [6]. Enzymes in the soil are useful biological indicators due to their association with soil ecology, sensitivity, operationally practicality, ease of evaluation, and integrative [3]. Enzymes come from many different soil sources, including non-living and living microorganisms, residues and roots of plants, and earth animals [7]. Activities of soil enzymes are more reliable and sensitive bioindicators than many other indicators that include plants and animals of any natural and anthropogenic disruption. They quickly respond to induced changes. On the other hand, enzyme activities can be influenced by unknown natural and anthropogenic activities, either in a significant or minor amount [3].

The difference in soil enzyme activity is due to the variation in soil organic matter, microbial community, microbial activity related to soil biology processes affected by both the abiotic and biotic factors [8]. The urease and invertase enzyme activity are essential indicators of microbial mediated functions and activities due to its rapid changes towards significant environmental changes [9]. Enzymes are the critical effectors of all the alterations happening in the ecology. They also catalyze with both narrow (chemo-, region- and stereo-selectivity) and broad specificity. They tend to perform processes for which no efficient chemical transformations have been developed [10]. Enzyme activity (EA) accelerates the reaction of the rate-limiting steps for organic matter degradation. Many studies have observed relationships between the plant litter decomposition, the microbial community, and enzyme activity. Naturally, enzyme reactions are sensitive towards the temperature, climatic, and geographic factors that can affect enzyme activity by changing the microbial biomass and the abiotic control of enzyme turnover and stabilization [11].

Climate can significantly affect the soil enzyme activity because of their sensitivity to temperature and precipitation. Other studies indicate that some areas with harsh climatic conditions exhibit less waste input, lower decomposability, and reduced numbers of microorganisms and enzyme activity [11]. Both the production and turnover rates of enzymes might be affected by temperature and moisture content; hence they may differ seasonally and be affected by climate change [12]. A study by Zheng et al., 2017 reported that soil pH has substantial effects on the structure and diversity of soil bacterial communities, and a suitable pH can benefit microbial growth. They said that both climatic situations and soil pH are the primary factors affecting soil enzyme activity.

Furthermore, the outcomes revealed that most analysed enzyme activity decreased with soil depth, but showed an increase in soil organic matter [11]. One of the essential soil components is soil organic carbon and organic matter because they can maintain soil fertility and crop production and prevent soil degradation, erosion, and desertification. A vital role in the decomposition of organic matter played by soil enzymes [13].

Most enzymes are sensitive to pH. They have a specific pH range and optimum pH of activity [14]. Many enzymes are important for catalysing different significant reactions necessary for soil for microorganism’s life processes and the soil structure maintenance, for the breaking down of organic wastes, organic matter formation, and nutrient cycling [6]. The activity of enzymes can be used to discover the level of pollution in soil (such as heavy metals, SO_4_), assess the successional stages, and degrade pesticides in the soil environment [7].

Several researchers have noticed the significant reasons why soil enzyme activity used as an essential soil-quality indicator. They noted that enzyme activity is often closely related to critical soil-quality parameters such as organic matter, soil physical properties, microbial activity, or biomass. Also, they can begin to change much sooner (1–2 years) than other properties (e.g., soil organic carbon); thus, they provide an early indication of the course of soil quality with changes in soil management. They can be an integrative soil biological index of past soil management [15].

Temperature and moisture can affect both the overall rate of enzyme production and the relative quality of production of different enzymes because they influence enzyme efficiency, substrate availability, and microbial efficiency. Therefore, alteration in the soil microclimate, whether they occur within hours, weeks, seasonally, or over decades as feedback to climate change, will influence the enzyme pool [12]. The aim of this study is to examine the activity of enzymes (urease, invertase, phosphatase, and catalase) in both polluted and unpolluted soils, to explore the relationship between enzyme activity and soil physicochemical properties as well, in order to identify the suitable active enzymes that can be used to monitor and manage the soil pollution.

## 2. Materials and Methods 

### 2.1. Description of the Study Area

A study had carried out in Alice Township under the Nkonkobe Municipality situated along the southern slopes of the Winterberg Mountains range and escarpment in the Province of the Eastern Cape. The sample sites are the Alice dumpsite and East campus inside the University of Fort Hare. The dumpsite lies between the latitudes of 32°48′24.88″ S and longitudes of 26°49′33.37″; E, whereas the control site lies between the latitudes of 32°47′07.35″ S and longitudes of 26°57′26.10″ E, respectively. The landfill site is located about 2 km from the Happy Rest residential area. The East campus was used as a control site and is ≈ 4 km from the dumping site.

The control site (that is Site 2) found a way down in the bottom of Somgxada hills that lay alongside the University fencing. The soil is protected by abundant natural vegetation. Site 1, which is the landfill site has divided into three portions, namely A, B, and C. Portion A is found on the east side of the dumpsite, where the ground covered by plenty of rusted and burned tins, broken bottles, or glasses, and rusty wires from car tires. While portion B is situated where the trucks and motor vehicles deliver garbage that is the west side. Last, portion C is positioned outside the dumpsite fencing, and many different natural plants covering the surface. 

#### 2.1.1. Collection and Preparation of Soil Sample

The soil samples were randomly collected at a depth of 0–25 cm every week for ten weeks. The sampling was carried out in sites 1and 2 using the clean soil auger. The dry soil samples collected were entirely placed in clean, labelled polyethylene bags and then transported to the laboratory for further analysis [16]. The soil samples were grinded using mortar and pestle to reduce the particle size and then sieved through a 2 mm mesh to obtain acceptable and homogeneous samples. Each sample was divided into two, and the first part stored at room temperature until the physicochemical analysis was performed. Simultaneously, the other part was stored in the refrigerator at four (4) °C until enzyme analyses were performed [17].

#### 2.1.2. Determination of the Physicochemical Properties of Soil

##### Soil Moisture Content

In a beaker of known mass, the soil samples were added and measured the mass. The samples were oven-dried at 105 °C for 24 h until the constant mass acquired. After drying, the samples were cooled in a desiccator, and the final mass was measured [18]: (1)$$\mathrm{Calculation}:\;\%\;\mathrm{Mass}\;=\;\left(\frac{\mathrm{Air}\;\mathrm{dried}\;-\;\mathrm{Oven}\;\mathrm{dried}}{\mathrm{Oven}\;\mathrm{dried}}\right)\times 100\;\%.$$

##### Soil pH

In a beaker, 1.0 g of a soil sample from each site was measured and stirred with 10 mL of deionized water (1:10 soil: distilled water mixture). Before measuring the pH, the digital pH meter calibrated using a standard buffer solution at the pH value of 4.01 and 7.00. The pH readings were taken by immersing the glass electrode/probe into the solution and record the reading [19].

##### Soil Electrical Conductivity (Salinity)

A 10 mL of well-mixed water sample was added into a measuring cylinder. A previously prepared soil sample was added to the water until the container’s contents increase by 5 mL to bring the volume to 15 mL. More water was added to the mixture to bring the total volume to 30 mL. The content was shaken intermittently for 5 min and allowed to settle for 5 min. An EC probe was dipped into the solution to measure the electrical conductivity [20].

##### Soil Organic Carbon

The 1 g soil sample has been treated with 5 mL of concentrated H_2_SO_4_ for 4 h, then with 5 mL of 0.5 M K_2_Cr_2_O_7_. The mixture was heated at 150–160 °C for 5 min and then cooled at room temperature. The solution moved into a conical flask with 100 mL deionized water. The unreacted K_2_Cr_2_O_7_ was determined by titrating with 0.25 M FeSO_4_. The endpoint had approached when the solution changes from dark green colour to blue to reddish-brown colour [19]:(2)$$\;\mathrm{Organic}\;\mathrm{carbon}\;\left(\%\right)\;=\;M\;\times \;\left(\frac{V1-\;V2}{\mathrm{Mass}\;\mathrm{of}\;\mathrm{soil}}\right)\;\times \;0.39.$$
where M = concentration of FeSO_4_, V1 = Volume of blank, V2 = Volume of FeSO_4_, 0.39 = constant.

### 2.2. Enzyme Activity Assays

#### 2.2.1. Invertase Enzyme

A 5 g of soil sample had mixed with 15 mL of 8% sucrose solution, 5 mL of distilled water, and five drops of toluene. The solution has incubated for 24 h at 37 °C. The solution was then centrifuged at 4000 rpm for 5 min, and a 1 mL aliquot moved into a volumetric flask containing 3 mL of 3,5dinitrosalicylic acid. The solution was heated for 5 min and cooled at room temperature. The content of glucose was quantified colorimetrically at 508 nm on a spectrophotometer. The invertase activity is expressed as μgglucose g^−1^soil h^−1^ [5]. 

#### 2.2.2. Urease Activity

A soil sample of about 5 g had mixed with 5 mL of toluene, 20 mL of distilled water, and 10 mL of 10% urea solution. The mixture was incubated at 37 °C for 24 h. After incubation, the solution was centrifuged for 5 min at 4000 rpm, 1 mL aliquot was mixed with 4 mL of sodium phenol solution (containing 100 mL of 6.6 M phenol solution and 100 mL of 6.8 M NaOH) and 3 mL of 0.9% sodium hypochlorite solution. The released ammonium in the solution was quantified colorimetrically at 578 nm on a spectrophotometer, and the urease activity was expressed as μgNH4N g^−1^ soil h^−1^ [5].

#### 2.2.3. Phosphatase Activity

The reaction mixture of 5 g soil, five drops of toluene, 10 mL of distilled water, and 10 mL of disodium phenyl phosphate solution had used to analyse phosphatase activity. The suspension was incubated at 37 °C for 24 h and centrifuged for 5 min at 4000 rpm. To add colour to the supernatant, a 0.25 mL of ammonia-ammonium chloride buffer, at pH 9.6, 0.5 mL of 2 % 4-amino antipyrine, and 0.5 mL of 8% potassium ferrocyanide was added to the solution. The phenol content was determined colorimetrically at 510 nm on a spectrophotometer. Phosphatase activity was expressed as μgphenol g^−1^soil h^−1^ [5]. 

#### 2.2.4. Catalase Activity

A 5 g of soil sample had mixed with 0.5 mL of toluene into a conical flask. The solution was stored at 4 °C for 30 min, after which a 5 mL of 3% H_2_O_2_ solution was added and stored again at 4 °C for 1 h. Afterward, 2 M of H_2_SO_4_ was added to the solution, and then the suspension was titrated with 0.01 M of KMnO4 until a faint pink colour appears in the solution. Catalase enzyme activity was expressed as mL of KMnO4 g^−1^soil h^−1^ [5].

### 2.3. Statistical Analysis

The data were analysed using the IBM Statistical Package for Social Science (SPSS) 25, version 25.0 (IBM, Armonk, NY, USA). Tukey Post-hoc tests at *p* ≤ 0.05 determined the multiple comparisons of means from one-way analysis of variance (ANOVA) and the significant difference among the selected enzyme activity means. IBM Pearson’s correlation analysed the relationships between soil enzyme activity and physicochemical parameters.

This study was approved for ethical clearance by AREC University of Fort Hare, with certificate number: OYE021SMAP01/19/E. 

## 3. Results

### 3.1. Soil Physical Properties

The soil physicochemical properties of 20 soil samples are shown in Table 1. Soil physicochemical properties are known to be the fundamental indicators for estimating the level of soil nutrient contents and characteristics. Soil electrical conductivity can serve as a measurement of soluble nutrients, and it is useful in monitoring the mineralization of organic matter in the soil. The availability of water content and soil pH in soil appears to be significant limitations of enzyme activity. Our study showed that soil enzymatic activities of urease, catalase, invertase, and phosphatase are affected directly one way or the other by levels of soil physicochemical parameters, like moisture content, pH, organic carbon, and electrical conductivity.

The moisture content results are introduced in Table 1 and Figure 1. The mean concentration values of moisture content were detected to range from 0.12 ± 0.07% to 9.09 ± 5.25%. The highest concentration levels for moisture content are recorded within the second week of sampling (with the mean of 6.42 ± 2.93%), followed by the week 1 sample with a mean value of 3.99± 1.40%; the minimum values are discovered in the fifth week. The dumpsite (site 1) soil samples were noted to contain high moisture content percentage than the samples collected on the control site (site 2). Both site 1B and Site 2 are allocated at the land’s vertical slope (sharp downward slope), where fewer plants cover the surface.

Soil pH is a measure of the acidity or alkalinity of soil [21]. The mean concentration of soil pH in four collection sites was analyzed, and we noted that the values range from slightly acidic (5.8 ± 3.36) to 8.51 ± 4.71) slightly alkaline soil (Table 1). The Tukey post-hoc analysis compares the soil pH values between the weeks, and the obtained data shows that the lowest mean values of soil pH reported on week 5 (6.11 ± 0.23), and maximum mean values seen at week 4 (8.27 ± 0.24). There was a remarkable random variation among the soil pH values detected between the collection areas. The standard permissible limits from WHO for pH values in the soil are required to range between 6 and 8.5 [22].

The soil electrical conductivity had measured in all collected soil samples to define the possible effects of salinity. The mean values for electrical conductivity were recorded to range between 2.06 ± 1.19 b µs/cm to 1540 ± 889.12 µs/cm (as presented in Table 1 and Figure 2). The EC mean values were noted to differ from one collection point to another randomly. The maximum values of electrical conductivity recorded in the first week of the collection, with the highest mean value of 1540 ± 889.12 µs/cm on-site A1. In contrast, a small amount was reported at site 1B. The Tukey post-hoc results confirm and indicate that week 1 consists of high EC values of 606.70 ± 698.17 µs/cm. The lowest values were observed in the second week, with a mean value of 322.10 ± 221.85 µs/cm.

The presence of organic carbon in soil was discovered, the mean percentage of soil organic carbon in the study has been noted to range between the value of 0.08 ± 0.05% and 1.54 ± 0.89% (Table 1). The mean values of soil organic carbon were observed to vary significantly from one area site to another, and values randomly differ within the site; there is no consistent sequence. The post-hoc multiple comparison test revealed that minimum organic carbon levels were recorded in the fourth week with the mean value of 0.42 ± 0.19%, and the high levels were observed in the first week of collection sites with a value of 0.98 ± 0.44%.

### 3.2. Soil Enzyme Activity Results

Both Table 2 and Figure 3 show the results of soil enzyme activity. In this study, we noted that the urease activity is deficient from all collection sites, especially on Site 1A with a mean concentration of 0.15 μg NH4-N g^−1^ soil h^−1^. Even though the urease activity was detected below the limit, the control site (Site 2) samples were noted to contain more (0.28 μg NH4-N g^−1^ soil h^−1^) urease activity more than the other three sampling sites.

The invertase activity was noted to be the most detected enzyme in all sites. It was recorded in high amounts in all four collection sites. The soil from Site 1C recorded high invertase activity 1.66 μg glucose g^−1^ soil h^−1^, followed by Site 2 with 1.64 μg glucose g^−1^ soil h^−1^, and the minimum invertase activity was observed at Site 1A (1.60 μg glucose g^−1^ soil h^−1^). The control site samples were noted to contain low levels of catalase activity (0.37 mL KMnO4 g^−1^ soil h^−1^), while Site 1A soil showed to have high catalase activity of 0.65 mL KMnO4 g^−1^ soil h^−1^. The high phosphatase activity levels were observed on soils from Site 1C (1.64 μg phenol g^−1^ soil h^−1^), followed by Site 1B samples (1.49 μg phenol g^−1^ soil h^−1^), and the minimum levels recorded at Site 1A (1.21 μg phenol g^−1^ soil h^−1^). There was no consistent trend detected among the soil enzyme activity. Most of the soil enzyme activity appeared higher on Site 1C samples, except for the catalase enzyme activity, which showed a different trend and was recorded high in Site 1A.

### 3.3. Correlation Results

The Pearson correlation coefficient was conducted, and the results show the positive but non-significant correlation between the soil physicochemical properties and selected enzymes activities (urease, invertase, catalase, and phosphatase) in soil, except for moisture content that correlated negatively with urease activity (as shown on Table 3). The moisture content correlated significantly and positively with the activity of invertase and catalase enzyme. Conversely, the invertase activity showed a negative correlation with soil pH, electrical conductivity, and organic carbon content. Soil pH had a significant negative but direct effect on phosphatase activity. A positive correlation was observed between the activity of catalase and soil electrical conductivity (EC). However, there was no significant relationship with other physicochemical properties.

## 4. Discussion

Water occurs and available naturally on soils and profoundly affects soil behaviour and the degree of compaction on the earth. In the present study, we noticed that oven drying test data reveals that moisture content is in a very low percentage of <10%, and it was noticed to range between 0.12 ± 0.07% on the fifth week (site 2) and 9.09 ± 5.25% on the second week (at site 1A). The polluted soil samples were reported to have higher moisture content than the control site (site 2). The low levels at site B might be attributed to the land’s vertical slope where the site was allocated, while on-site 1 moisture might come from some disposed garbage. Furthermore, this study’s moisture content levels were generally very low because of the lack of rainfall during the collection period (climate conditions and seasonal changes). The ground surface in these sites tends to evenly downwards and might results in low moisture content because water is hardly absorbed on the surface during the runoff or rainy days. Low water levels in soil (drying of soil) can decrease or complete soil enzyme activity [23]. Our experimental study shows that low moisture content levels directly affected the activity of enzymes in soil, confirmed by minimal amounts of enzyme activity detected in this study.

Due to low moisture content, the activity of enzymes in soil was also affected, resulting in poor soil health and lack of soil nutrients for plant uptake since there will be deficient nutrients recycled by these enzymes. The study reveals that moisture content is a remarkably important edaphic factor that changes soil’s biological activity. A study by Landesman and Dighton 2010 agrees with our study’s findings. They reported that dry weather conditions where there is low moisture lead to reducing microorganism’s biomass [24]. Another study by Borowik and Wyszkowska in 2016 reported that the highest enzyme activities in soils were obtained with the moisture content above 20%. High moisture content in soil enhances the development of microorganisms and reflects the oxygen conditions [25], increases enzyme activity and the production of organic matter in the form of organic carbon [26].

The soil pH is the primary factor that affects the mobility and the solubility of metals in the soil environment [27]. The soil pH testing outcomes reveal that the sample’s pH ranges from slightly acidic conditions (5.8 ± 3.36) to moderately alkaline soils (8.51 ± 4.91). The obtained soil pH has been noted to falls within the normal range set by WHO for both sites, except for one soil sample collected at site 1B, which was recorded slightly below the allowable limits. It has been reported by Neina 2019 that enzymes mostly become more active between pH of 6.8 to 8.0 [28]. In this study, the results disagree with the written finding because, in this study, enzyme activity levels were detected in very small amounts and ranged below 5%. Another study said that at 6.5 pH, the highest levels of enzyme activity were identified [29].

The movement and the availability of macronutrients and micronutrients in the soil are affected by pH and other soil components. As the pH decreased, the solubility of metallic components in soil rose and became more freely available in different fractions because of the rise in ions’ solubility in acidic surroundings. Based on this study’s results, we can conclude by saying both micro and macronutrient availability will be increased, their solubility and mobility will also be enhanced, and beneficial soil organisms will be most active [30].

Electrical conductivity is a very liable test for soil salinity [21]. It represented the conduction current ability of soil solution and sustained ion concentration increase under electric field effects that leads to conductivity increase [31]. The salinity test on this study reveals that soil samples have a low electrical conductivity that’s is noticed to range between 1540 ± 889.12 μs/cm and 2.06 ± 1.19 μs/cm. Low electrical conductivity in this study shows that they can’t conduct electrical current and affect plant yield and health.

The electrical conductivity is temperature dependant; if the temperature increases, electrical conductivity also increases [32]. Therefore, low levels of electrical conductivity in this study might be due to low-temperature levels during the sampling period (which was a winter season). As presented in Table 1, the results show that the soil from the landfill site (site 1) contains high EC values compared to the soil from the control site (site 2). Electrical conductivity is used mostly in the estimation of soluble salts concentration. The levels of EC that are below 0–2 Ms/cm are marginally or non-saline soil. In this study, the analysed soil samples were recorded within the range of non-saline soil. The reported findings by Lemanowicz et al. 2018 contradict the results of this study. They said that high electrical conductivity decreases the enzymatic activity on soils, while in our study, we detected low electrical conductivity levels and very low enzyme activity recorded [33]. The variation in mean values of electrical conductivity might be due to differences in soluble salts content in samples [23]. The topography is observed to positively influence many soil properties that include electrical conductivity [34].

The organic carbon in soil is a crucial determinant for soil quality and productivity. There is a close relationship between soil enzyme activity and soil organic carbon, mainly affected by horizons and other factors [35]. We have examined a shallow content of soil organic carbon below 2%, and it ranges in a small percentage of 0.08 ± 0.05% to 1.54 ± 1.19%. We can report that soil organic carbon directly affects enzymes’ activity due to the low enzyme activity observed. The study by Zhao et al. 2018 reported similar findings, and that supports our results. The maintenance of soil organic carbon is the main factor favouring soil microbial biomass and enzyme activity [36]. Overall the percentage of SOC (soil organic carbon) in both sampling sites were recorded in low values; this might be attributed to reduced root biomass in the sampling site, which may decrease the labile (carbon) C input to mineral soil in the form of root excretions. Also, in this study, the moisture content had recorded in a low amount, so low moisture in soil limits C’s downward movement from waste materials to soil, resulting in the reduction of organic carbon in soil [37].

High levels of soil enzyme activities were recorded on Site 1 samples, with invertase activity (1.66 μg glucose) g^−1^ soil h^−1^and phosphatase activity (1.64 μg phenol g^−1^ soil h^−1^) recorded high at Site 1C and Site 1B. Invertase and phosphatase activity appeared to be suitable enzymes for monitoring soil pollution. Their activity might be used as bioindicators of soil health, soil quality, and soil productivity. The study by Maharana and Patel (2013) reported similar findings, where the phosphatase enzyme activity was recorded beyond the datable limit on all the sampling sites and noted a significant positive correlation between phosphatase activity and soil physicochemical properties [38].

On the contrary, Chineyre et al. 2013 reported that highly polluted dumpsite soil contains significantly high urease activity (*p* ≤ 0.05) because this enzyme originates mainly from plants and microorganisms. Its high concentration is also associated with the household waste materials dumped, which contains urea [39]. Urease enzyme is essential, and it signifies 63% of total enzyme activity in the soil. The urease activity variation is instigated by modification in soil physicochemical parameters such as the content of moisture and organic matter and the accumulation of nitrogen, which is regarded as the urease substrate in soil. Kumari et al., 2017 also noted a non-significant correlation between urease activity and soil pH [40].

Low levels of enzymatic activity in this study show the critical role of water content for microbial enzyme activity in soil [41]. Low urease enzyme activity levels in the soil might be linked with low organic carbon levels on an excellent decomposition rate, confirmed by low levels of organic carbon observed in this study [4]. The urease enzyme is known to originate from plants and microorganisms. Therefore, low urease enzyme activity in our study might occur as the effects of deficient plant availability in soil due to the plant disturbance and seasonal changes [5]. Plant coverage in soil affects the soil properties directly and indirectly by modifying the microclimate conditions [7].

Urease activity is revealed to positively connect with the total nitrogen (N) in soil. Hence, low urease activity levels in the soil might occur as the result of the decrease in total N on soil samples [42]. Since the incineration method is used in dumpsites to decrease waste materials, it reduces urease activity levels due to the fire effects on soil [3]. The urease activity was very low in dumpsites due to the fire effect. The waste on the site is being reduced or destroyed through incineration method, leading to intense urease enzyme activities in soil. Low levels of soil enzyme activities in our study could be because of seasonal changes. The samples were collected in the spring season, and the spring season is known for cold weather and no rain conditions. The burning of waste in the dumpsites results in the decrease of soil microorganisms and biomass, which directly impacts the availability and the activity of soil enzymes. This study also shows that enzyme activity is closely related to the soil organic carbon and moisture content in the soil. Moreover, enzyme activity can be considered a useful indicator of soil quality changes from environmental disturbances and management practices.

Also, during the sampling period for this study, the temperature was low. It was detected to be below 20 °C, resulting in low levels of urease activity in soil samples. The study by Wang et al., 2016 supports our findings. It confirms that during the spring season, the urease, invertase, and catalase activity are recorded at low values compared to other seasons of the year [43]. Similarly, Utobo and Tewari (2015) study show the effects of the temperature of enzyme activity. They noted that generally, the urease activities increase with the increase in temperature [3]. Another study by Mondal et al., 2015 agrees with the previously reported research that urease activity increases with the increasing temperature from 10 °C to a maximum at 60 °C. A further increase in temperature decreased urease activity, which virtually inhibited at 100 °C. As the temperature changes with the season, seasonality in enzyme activity is justified. Seasonal changes in the microorganisms that produce these enzymes may also give rise to seasonality [44].

The descending order for all selected soil enzyme activities in collection sites were recorded as follows invertase activity > phosphatase activity > catalase activity > urease activity. The one-way ANOVA analysis (as shown in Appendix A) was conducted to compare different enzymes’ activity levels in polluted soil samples. The results reported as means (M) and standard deviation (SD), the obtained results revealed that there is a statistically significant difference among the four enzymes analysed, where F (3, 156) = 76.002, and *p* ≤ 0.05 = 0.001. The Tukey post- hoc testing also revealed significant differences between the groups of enzyme activities, where urease activity (M = 0.213, SD = 0.357) and catalase activity (M = 0.492, SD = 0.409) are having low levels of enzyme activity when compared to the activity of invertase (M = 1.599, SD = 0.574) and Phosphatase PHO activity (M = 1.410, SD = 0.590). These results reveal that the invertase and phosphatase enzymes had higher activity in contaminated and uncontaminated soils than the urease and catalase enzymes. This shows that the urease enzyme’s mean activity is significantly different from the mean activity of invertase and phosphatase, while there was no significant difference with the catalase enzyme.

For the correlation analysis even though the soil EC showed a direct positive effect and significant relationship on catalase activity, a regression model revealed that soil EC and other soil physicochemical properties except for soil total P were insignificant in reflecting the quantitative connection catalase activity [45]. A significant correlation between the catalase activity and soil organic carbon was reported in this study, which agrees with Zhao et al., 2018. Soil organic carbon is noted to be a positive factor in soil quality. Hence low levels of organic carbon reveal poor soil quality [36]. Tan et al. 2014 reported different findings from this study, and they noted a non-significant correlation between soil pH and phosphatase activity (*p* > 0.05). However, soil pH negatively correlated with invertase, urease, and catalase activities (*p* < 0.01) [5]. Another study reported that other soil properties such as soil pH, available P, and electrical conductivity (EC) do not affect urease activity.

## 5. Conclusions

In this study, four soil enzyme activities were successfully identified, and the effects of soil physicochemical properties on enzymes were reported. Invertase and phosphatase enzyme activity seem to be the appropriate enzymes for monitoring soil pollution and soil management due to their high activity level recorded for this study. The phosphatase enzyme activity correlates positively with all detected physicochemical parameters (moisture content, pH, organic carbon, and electrical conductivity), while the invertase activity showed a negative correlation towards physicochemical properties for moisture content in which they significantly correlated. For this study, we found that the variability in SOC content is also affected by different sampling sites, mainly because various sites differ in physicochemical characteristics and enzyme activities. We have noticed that enzymes are poorly available, leading to low soil quality and environmental health. Enzymes in the soil are suppressed, which results in the prevention of essential processes that can affect the cycling of nutrients for plant growth and sustainability. Insufficient enzyme activity can result in an accumulation of harmful chemicals to the environment; some of these chemicals may further inhibit soil enzyme activity. The results show that there is a need for the implementation of waste management programs, which will help the municipality to prioritize issues of illegal dumping so that there will be an improvement in economic growth, living conditions of humans in affected areas, and reduction of unfavourable influence on the soil environment. This study also recommends evaluating soil enzyme activity in soil as it is one of the cheapest and easiest practices that can be used to assess soil pollution. Their evaluation will provide important information on soil microbial activity. It will help the authorities examine the degree of soil pollution by heavy metals and maintain soil quality.

## Figures and Tables

**Figure 1 ijerph-18-00221-f001:**
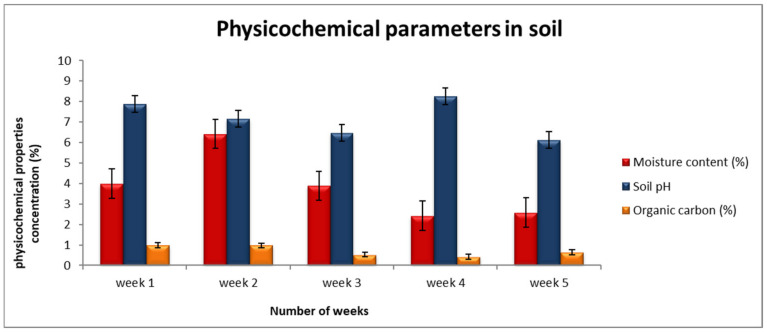
The soil physicochemical properties result for the five weeks sampling period.

**Figure 2 ijerph-18-00221-f002:**
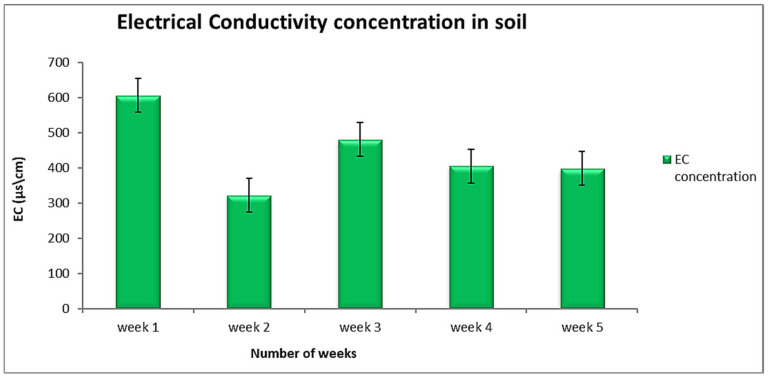
The electrical conductivity results in the soil for the five weeks sampling period.

**Figure 3 ijerph-18-00221-f003:**
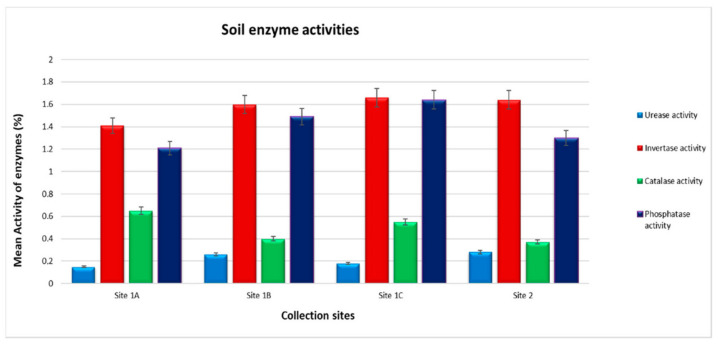
Selected soil enzyme activities in four collection sites.

**Table 1 ijerph-18-00221-t001:** Results of Physicochemical properties in soil samples.

		Moisture Content (%)	pH	Electrical Conductivity (μs/cm)	Organic Carbon (%)
Week 1	Site 1A	2.24 ± 1.29 *^a^*	7.65 ± 4.42 *^a^*	1540 ± 889.12 *^a^*	1.13 ± 0.65 *^a^*
	Site 1B	4.29 ± 2.48 *^b^*	7.67 ± 4.43 *^a^*	2.06 ± 1.19 *^b^*	0.58 ± 0.33 *^b^*
	Site 1C	5.64 ± 3.26 *^ab^*	7.86 ± 4.54 *^a^*	735.33 ± 424.54 *^ab^*	1.54 ± 0.89 *^a^*
	Site 2	3.79 ± 2.19 *^c^*	8.29 ± 4.79 *^b^*	149.40 ± 86.26 *^c^*	0.68 ± 0.39 *^b^*
Week 2	Site 1A	9.09 ± 5.25 *^bc^*	7.11 ±4.10 *^ab^*	567 ± 327.36 *^cd^*	0.35 ±0.20 *^ab^*
	Site 1B	4.30 ± 2.48 *^b^*	6.89 ± 3.98 *^c^*	391 ± 225.74 *^d^*	1.43 ± 0.83 *^a^*
	Site 1C	8.79 ± 5.07 *^bc^*	7.22 ±4.17 *^ab^*	295 ± 170.32 *^d^*	0.48 ±0.28 *^ab^*
	Site 2	3.50 ± 2.02 *^c^*	7.36 ±4.25 *^ab^*	35.4 ± 20.44 *^b^*	0.55 ± 0.32 *^b^*
Week 3	Site 1A	4.80 ± 2.77 *^b^*	6.92 ± 3.99 *^c^*	327 ± 188.79 *^d^*	0.74 ±0.43 *^b^*
	Site 1B	3.40 ± 1.96 *^c^*	7.20 ±4.16 *^ab^*	86.3 ± 49.82 *^b^*	0.46 ±0.27 *^ab^*
	Site 1C	3.81 ± 2.20 *^c^*	5.61 ±3.24 *^bc^*	1445 ± 834.27 *^a^*	0.65 ± 0.37 *^b^*
	Site 2	3.54 ± 2.04 *^c^*	6.18 ±3.57 *^bc^*	67.5 ± 38.97 *^b^*	0.26 ±0.15 *^ab^*
Week 4	Site 1A	4.02 ± 2.32 *^b^*	7.95 ± 4.59 *^a^*	691 ± 398.95 *^ab^*	0.44± 0.25 *^ab^*
	Site 1B	1.18 ± 0.68 *^d^*	8.51 ±4.91 *^b^*	306 ± 176.67 *^d^*	0.48± 0.28 *^ab^*
	Site 1C	2.81 ± 1.62 *^a^*	8.40 ± 4.85 *^b^*	77.1 ± 44.51 *^b^*	0.60 ± 0.35 *^b^*
	Site 2	1.69 ± 0.98 *^d^*	8.22 ± 4.75 *^b^*	549 ± 316.96 *^cd^*	0.15 ± 0.09 *^c^*
Week 5	Site 1A	3.95 ± 2.28 *^c^*	6.07± 3.50 *^bc^*	741 ± 427.82 *^ab^*	1.35 ± 0.78 *^a^*
	Site 1B	1.36 ± 0.78 *^d^*	5.8 ± 3.36 *^bc^*	296 ± 170.90 *^d^*	0.53 ± 0.31 *^b^*
	Site 1C	4.91 ± 2.83 *^b^*	6.35± 3.67 *^bc^*	463 ± 267.31 *^cd^*	0.60 ±0.35 *^b^*
	Site 2	0.12 ± 0.07 *^cd^*	6.21± 3.58 *^bc^*	94.8 ± 54.73 *^b^*	0.08 ±0.05 *^c^*

Results are presented as mean values ± SD Means with different letters within the same column shows a significant difference (*p* < 0.05) at 95% interval. Letter *a*, *b*, *c* and *d* in the means show that there is a statistical significant difference between the variables in the column.

**Table 2 ijerph-18-00221-t002:** The results of enzyme assay in soil samples.

		Urease (μg NH4- N g^−1^ Soil h^−1^)	Invertase (μg Glucose g^−1^ Soil h^−1^)	Catalase (mL KMnO4 g^−1^ Soil h^−1^)	Phosphatase (μg Phenol g^−1^ Soil h^−1^)
Week 1	Site 1A	0.50 ± 0.29 *^a^*	1.35 ± 0.78 *^a^*	1.22 ± 0.70 *^a^*	2.85 ± 1.64 *^a^*
	Site 1B	1.96 ± 1.13 *^b^*	1.00 ± 0.58 *^a^*	0.55 ± 0.32 *^b^*	1.72 ± 0.99 *^b^*
	Site 1C	0.82 ± 0.47 *^a^*	0.18 ± 0.11 *^b^*	1.01 ± 0.58 *^a^*	3.49 ± 2.01 *^ab^*
	Site 2	3.21 ± 1.85 *^ab^*	1.08 ± 0.62 *^a^*	0.65 ± 0.37 *^b^*	2.22 ± 1.28 *^c^*
Week 2	Site 1A	0.15 ± 0.08 *^c^*	0.60 ± 0.34 *^ab^*	0.51 ± 0.29 *^b^*	0.55 ± 0.32 *^cd^*
	Site 1B	0.08 ± 0.04 *^c^*	1.31 ± 0.76 *^a^*	0.27 ± 0.16 *^ab^*	1.07 ± 0.62 *^d^*
	Site 1C	0.10 ± 0.05 *^c^*	2.28 ± 1.31 *^c^*	0.35 ± 0.20 *^ab^*	0.67 ± 0.38 *^cd^*
	Site 2	0.06 ± 0.03 *^c^*	0.79 ± 0.46 *^ab^*	0.15 ± 0.09 *^c^*	0.78 ± 0.45 *^cd^*
Week 3	Site 1A	0.20 ± 0.12 *^cd^*	1.85 ± 1.07 *^cd^*	0.21 ± 0.12 *^ab^*	1.04 ± 0.60 *^d^*
	Site 1B	0.98 ± 0.56 *^a^*	2.32 ± 1.34 *^c^*	0.32 ± 0.18 *^ab^*	1.19 ± 0.69 *^d^*
	Site 1C	1.44 ± 0.83 *^a^*	2.75 ± 1.89 *^c^*	0.25 ± 0.14 *^ab^*	2.38 ± 1.37 *^c^*
	Site 2	0.82 ± 0.47 *^a^*	2.35 ± 1.36 *^c^*	0.17 ± 0.10 *^c^*	1.27 ± 0.73 *^d^*
Week 4	Site 1A	0.99 ± 0.57 *^a^*	3.48 ± 2.01 *^d^*	1.45 ± 0.84 *^a^*	2.41 ± 1.39 *^c^*
	Site 1B	0.32 ± 0.18 *^cd^*	3.26 ± 1.88 *^d^*	0.43 ± 0.25 *^ab^*	2.82 ± 1.63 *^a^*
	Site 1C	0.10 ± 0.06 *^c^*	3.66 ± 2.11 *^d^*	0.89 ± 0.51 *^cd^*	2.76 ± 1.59 *^a^*
	Site 2	0.08 ± 0.05 *^c^*	3.48 ± 2.01 *^d^*	0.22 ± 0.13 *^ab^*	3.19 ± 1.84 *^ab^*
Week 5	Site 1A	0.13 ± 0.07 *^c^*	2.18 ± 1.26 *^c^*	0.75 ± 0.43 *^b^*	3.55 ± 2.05 *^ab^*
	Site 1B	0.18 ± 0.10 *^c^*	3.14 ± 1.81 *^d^*	0.35 ± 0.20 *^ab^*	2.55 ± 1.47 *^a^*
	Site 1C	0.05 ± 0.03 *^c^*	3.37 ± 1.94 *^d^*	0.43 ± 0.25 *^ab^*	3.37 ± 1.95 *^ab^*
	Site 2	0.07 ± 0.04 *^c^*	3.36 ± 1.91 *^d^*	0.28 ± 0.16 *^ab^*	3.85 ± 2.22 *^e^*

Results are presented as mean values ± SD; Means with different letters within the same column shows a significant difference (*p* < 0.05) at 95% interval. Letter *a*, *b*, *c* and *d* in the means show that there is a statistically significant difference between the variables in the column.

**Table 3 ijerph-18-00221-t003:** Correlation results between enzyme activity and physicochemical properties.

	Soil pH	Electrical Conductivity	Moisture Content	Organic Carbon
Urease	0.175	0.054	−0.297	0.021
Invertase	−0.070	−0.155	0.403 **	−0.329 *
Catalase	0.257	0.089	0.443 **	0.177
Phosphatase	0.128	0.082	0. 146	0.018

Note: ** Correlation is significant at the 0.01 level (2-tailed); * Correlation is significant at the 0.05 level (2-tailed).

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
