# Peer review of "The Effects of Physicochemical Parameters on Analysed Soil Enzyme Activity from Alice Landfill Site"

_ijerph, 2020, doi:10.3390/ijerph18010221_

Round 1
Reviewer 1 Report
Overall, the entire results section is poorly interpreted. It remains to be clearly presented where statistically significant differences were observed, with references to the respective test statistics and respective p-values.
MATERIAL AND METHODS
The statistical analysis can be improved and increased:
- In the material and methods, section of the "Statistical Analysis", the authors must complete the information with the indication of which variables were studied, which factor and whether the assumptions inherent to the analysis were validated;
- On the other hand, the version of the software that was used must be indicated (and it shouldn't be information at the beginning, but at the end of this section);
- Authors should clarify whether they performed the multiple comparison tests after performing the analysis of variance.
Results:
- In any scientific work, a section should never begin with a table. Thus, the authors have to reformulate section 3.1 and 3.3, referring to the results;
- The explanatory note in Table 1 and Table 2 is wrong. The authors report that: “Mean values with the same letters in a column are not significantly different 95% interval where P ˂ 0.05”, which is an inconsistency, since the results when they are not statistically significant present a p-value> 0.05. The authors have to correct and clarify this situation;
- The analysis and interpretation of the results cannot be presented based on descriptive statistical analysis (line 208). Given that, the authors refer in the "material and methods" section that they performed an analysis of variance, then the reason for analysing the data based on the described statistics is not understood. Clarification is requested;
- The same occurs on lines 219 and 226;
- On the other hand, the sentence “This was confirmed and supported by the descriptive statistical analysis (on ANOVA results)”, is completely wrong, since the analysis of variance is not a descriptive analysis, but an inferential analysis;
- When the authors refer: “The mean values of soil organic carbon were observed to vary significantly from one area site to another” (line 224), have to indicate which test of multiple comparisons used;
- In Figure 2, the situations in which statistically significant differences were observed should be noted;
- Given that there is a section dedicated to the discussion of results, then it will not be appropriate to have what appears at the end of the results section, namely “Tan et al.2014 reported different findings from this study, they noted a non-significant correlation that was found between soil pH and phosphatase activity (? > 0.05). However, soil pH negatively correlated with invertase, urease, and catalase activities (? < 0.01) [20]. Another study reported that other soil properties such as soil pH, available P, and electrical conductivity (EC) do not affect urease activity.”.
Author Response
Response to Reviewer 1 Comments
Point 1: In the material and methods, a section of the "Statistical Analysis", the authors must complete the information with the indication of which variables were studied, which factor, and whether the assumptions inherent to the analysis were validated.
Response 1: The variables studied were indicated (which is soil enzyme activity and physicochemical parameters) under the topic of Statistical analysis. The systemic factor (soil enzymatic activity) was used in this study, and the null hypothesis concept (statistical theory) was used to validate the made assumption on the obtained data.
Point 2: On the other hand, the version of the software that was used must be indicated (and it shouldn't be information at the beginning but the end of this section);
Response 2: The name of the software and its version used has been indicated in the statistical analysis section (IBM SPSS version 25.0).
Point 3: Authors should clarify whether they performed multiple comparison tests after performing the analysis of variance.
Response 3: Yes, after ANOVA analysis a Tukey Post- hoc multiple comparison tests were conducted.
Point 4: In any scientific work, a section should never begin with a table. Thus, the authors have to reformulate section 3.1 and 3.3, referring to the results
Response 4: Section 3.1 to 3.3 were restructured based on the results.
Point 5: The explanatory note in Table 1 and Table 2 is wrong. The authors report that: “Mean values with the same letters in a column are not significantly different 95% interval where P ˂ 0.05”, which is an inconsistency, since the results when they are not statistically significant present a p-value> 0.05. The authors have to correct and clarify this situation.
Response 5: On both Table 1 and Table 2 the explanatory note for means was reformulated and corrected.
Point 6: The analysis and interpretation of the results cannot be presented based on descriptive statistical analysis (line 208). Given that, the authors refer to the "material and methods" section that they performed an analysis of variance, then the reason for analysing the data based on the described statistics is not understood. Clarification is requested; The same occurs on lines 219 and 226;
Response 6: The interpretation of results has been restructured and presented using the multiple comparison results tested by Tukey post hoc. The ANOVA analysis was conducted to compare the means of soil enzymes and physicochemical properties to examine whether there is a shred of statistical evidence that the associated enzyme activity means are significantly different.
Point 7: On the other hand, the sentence “This was confirmed and supported by the descriptive statistical analysis (on ANOVA results)”, is completely wrong, since the analysis of variance is not descriptive, but an inferential analysis.
Response 7: The sentence on ANOVA results has been reformulated and rewritten in the correct form.
Point 8: When the authors refer: “The mean values of soil organic carbon were observed to vary significantly from one area site to another” (line 224), have to indicate which test of multiple comparisons used;
Response 8: The multiple comparison test that is used on soil organic carbon analysis indicated that it is a Tukey post – hoc test.
Point 9: In Figure 2, the situations in which statistically significant differences were observed should be noted;
Response 9: The statistical analysis on enzyme activity was observed and it was reported under the discussion section (line 398 to 409)
Point 10: Given that there is a section dedicated to the discussion of results, then it will not be appropriate to have what appears at the end of the results section, namely “Tan et al.2014 reported different findings from this study, they noted a non-significant correlation that was found between soil pH and phosphatase activity (? > 0.05). However, soil pH negatively correlated with invertase, urease, and catalase activities (? < 0.01) [20]. Another study reported that other soil properties such as soil pH, available P, and electrical conductivity (EC) do not affect urease activity.
Response 10: The paragraph “Tan et al.2014 reported different findings from this study, they noted a non-significant correlation that was found between soil pH and phosphatase activity (? > 0.05). However, soil pH negatively correlated with invertase, urease, and catalase activities (? < 0.01) [20]. Another study reported that other soil properties such as soil pH, available P, and electrical conductivity (EC) do not affect urease activity.” has been removed under results section and replaced under discussion section.
Reviewer 2 Report
This study aims to determine the activity of enzymes (urease, invertase,
phosphatase, and catalase) in both polluted and unpolluted soils, to examine the relationship between enzyme activity and soil physicochemical properties as well as to identify the suitable active enzymes that can be used to monitor and manage the soil pollution.
- The relationship between some soil physicochemical properties and enzymes is not clear or evident. The Discussion session seems to try to describe the relationship, but the paragraphs intended for this presentation of the relationship between soil properties and enzymes, fail to make this objective clear. There is an explanation about the importance of physicochemical properties, but it does not explain the influence of the researched enzymes. This is well observed in the first 3 paragraphs of the Discussion session. I suggest that the authors add their comments.
- The results of the analysis of the physicochemical properties of the soil are interesting, and they are in the tables. There is a lot of text in an attempt to explain the relationship between physicochemical properties and enzymes, but it is not evident what the authors find in their research. I suggest that the authors try to present the results of these enzyme influences on soil properties in a graphical form in order to enrich the scientific presentation, more than that, to make the finding clear.
Author Response
Response to Reviewer 2 Comments
Point 1: The relationship between some soil physicochemical properties and enzymes is not clear or evident. The Discussion session seems to try to describe the relationship, but the paragraphs intended for this presentation of the relationship between soil properties and enzymes, fail to make this objective clear. There is an explanation about the importance of physicochemical properties, but it does not explain the influence of the researched enzymes. This is well observed in the first 3 paragraphs of the Discussion session. I suggest that the authors add their comments.
Response 1: The first three paragraphs in the discussion were reformulated and the relationship between the soil enzyme activity and physicochemical parameters was also indicated as well as the influence of physicochemical properties on the studied enzymes was also reported in the discussion section.
Point 2: The results of the analysis of the physicochemical properties of the soil are interesting, and they are in the tables. There is a lot of text in an attempt to explain the relationship between physicochemical properties and enzymes, but it is not evident what the authors find in their research. I suggest that the authors try to present the results of these enzyme influences on soil properties in a graphical form to enrich the scientific presentation, more than that, to make the finding clear.
Response 2: The graphical form of physicochemical parameters results was presented under the results section, and the text to explain their relationship with enzyme activities in soil was restructured and reported on the section.
Reviewer 3 Report
The study aims to determine the activity of enzymes (urease, invertase, phosphatase, and catalase) in both polluted and unpolluted soils, to examine the relationship between enzyme activity and soil physicochemical properties as well as to identify the suitable active enzymes that can be used to monitor and manage the soil pollution. However, there are some questions:
- The innovation of the study is limited. The key findings of the study may help government administrator to prioritize issues of illegal dumping to improve economic growth. However, the key findings should be examined in other study areas.
- The study problems and objective are unclear both in Introduction and Conclusion part.
- There are grammar errors. In page 5, line 187, “Statistical analysis:” should be “Statistical analysis”.
- The study methods are not new. In this reason, the contribution of the study is limited and not strong.
Author Response
Response to Reviewer 3 Comments
Point 1: The innovation of the study is limited. The key findings of the study may help government administrator to prioritize issues of illegal dumping to improve economic growth. However, the key findings should be examined in other study areas.
Response 1: The key findings for this study were noticed and reported under the discussion section
Point 2: The study problems and objectives are unclear both in the Introduction and Conclusion part.
Response 2: Both the objectives and the problems of the study were revealed in the introduction section and conclusion.
Point 3: There are grammar errors. On page 5, line 187, “Statistical analysis:” should be “Statistical analysis”.
Response 3: Online 187 the grammatical error was corrected.
Point 4: The study methods are not new. For this reason, the contribution of the study is limited and not strong.
Response 4: The methods used in this study were extracted from the recent and latest journals.
Reviewer 4 Report
L11-17 This part is so long. You do not need introduce so many backgrounds here.
L22 24 hours or 24 h
L25: Do not use SD in the abstract.
L36: In this part, you should clearly introduce that why you chose there four enzymes to explore the scientific problem.
L120: the fig.1 should be deleted.
L124: why did you choose 0-25 cm soil samples? As we know, many differences in the soil 0-10 cm and 10-15 cm.
L138-143 Soil ph and conductivity should be merge and simplification, so many title here.
L229: why did you measure the index every week?
L232: significance analysis should be added in the fig.
L269-271: the discussion should be simplification. So many backgrounds introduction were written in this part (L279-280 ETC.). You should only introduce your mainly results and compare them with other researches.
Author Response
Response to Reviewer 4 Comments
Point 1: L11-17 This part is so long. You do not need to introduce so many backgrounds here.
Response 1: The paragraph was reformulated and restructured.
Point 2: L22 24 hours or 24 h.
Response 2: It's 24 hours for the enzyme assay method.
Point 3: L25: Do not use SD in the abstract.
Response 3: The standard deviation values were removed in the abstract section.
Point 4: L36: In this part, you should clearly introduce that why you chose there four enzymes to explore the scientific problem.
Response 4: In line 36 under the introduction section the explanation of why four enzymes were selected for the analysis was reported.
Point 5: L120: the fig.1 should be deleted.
Response 5: figure 1 was removed under the study area section.
Point 6: L124: why did you choose 0-25 cm soil samples? As we know, many differences in the soil 0-10 cm and 10-15 cm.
Response 6: The 0 -25 cm sampling was chosen because it deals with topsoil surfaces and is the most recommended by agronomic guidelines of sampling for accurate analysis. More information on: Vadas, P. A.; Mallarino, A. P.; McFarland, A. (2006). The importance of sampling depth when testing soils for their potential to supply phosphorus to surface runoff. Extension Fact Sheets, 1.
Point 7: L138-143 Soil pH. and conductivity should be merge and simplification, so many titles here.
Response 7: Under results, section titles were removed, and information was presented in the form of paragraphs for each physicochemical properties.
Point 8: L229: why did you measure the index every week?
Response 9: For the accuracy, and good quality analysis, also to examine the influence of disposed garbage on both soil physicochemical parameters and enzyme activity.
Point 9: L232: significance analysis should be added in the fig.
Response 9: All the statistical significance analysis for soil enzyme activity was added and presented in the discussion section.
Point 10: L269-271: the discussion should be simplification. So many backgrounds introduction were written in this part (L279-280 ETC.). You should only introduce your mainly results and compare them with other researches.
Response 10: The discussion section was reformulated, and the obtained results were reported as well as the comparison to other studies were included in this section
Round 2
Reviewer 3 Report
The manuscript can be accepted after made grammar revision totally.